# Exploring post acute rehabilitation service use and outcomes for working age stroke survivors (≤65 years) in Australia, UK and South East Asia: data from the international AVERT trial

Rosy Walters [1,2] Janice M Collier [2] Lillian Braighi Carvalho,[2]
Peter Langhorne,[3] Md Ali Katijjahbe,[4,5] Dawn Tan,[6] Marj Moodie,[7]
Julie Bernhardt [2] AVERT Trialists' Collaboration

For numbered affiliations see end of article.

**Correspondence to**
Dr Julie Bernhardt;
julie.bernhardt@florey.edu.au

## ABSTRACT

**Objectives** Information about younger people of working age (≤65 years), their post stroke outcomes and rehabilitation pathways can highlight areas for further research and service change. This paper describes: (1) baseline demographics; (2) post acute rehabilitation pathways; and (3) 12-month outcomes; disability, mobility, depression, quality of life, informal care and return to work of working age people across three geographic regions (Australasia (AUS), South East (SE) Asia and UK).

**Design** This post hoc descriptive exploration of data from the large international very early rehabilitation trial (A Very Early Rehabilitation Trial (AVERT)) examined the four common post acute rehabilitation pathways (inpatient rehabilitation, home with community rehabilitation, inpatient rehabilitation then community rehabilitation and home with no rehabilitation) experienced by participants in the 3 months post stroke and describes their 12-month outcomes.

**Setting** Hospital stroke units in AUS, UK and SE Asia.

**Participants** Patients who had an acute stroke recruited within 24 hours who were ≤65 years.

**Results** 668 participants were ≤65 years; 99% lived independently, and 88% no disability (modified Rankin Score (mRS)=0) prior to stroke. We had complete data for 12-month outcomes for n=631 (94%). The proportion receiving inpatient rehabilitation was higher in AUS than other regions (AUS 52%; UK 25%; SE Asia 23%), whereas the UK had higher community rehabilitation (UK 65%; AUS 61%; SE Asia 39%). At 12 months, 70% had no or little disability (mRS 0–2), 44% were depressed, 28% rated quality of life as poor or worse than death. For those working prior to stroke (n=228), only 57% had returned to work. A noteworthy number of working age survivors received no rehabilitation services within 3 months post stroke.

**Conclusions** There was considerable variation in rehabilitation pathways and post acute service use across the three regions. At 12 months, there were high rates of depression, poor quality of life and low rates of return to work.

**Trial registration number** Australian New Zealand Clinical Trials Registry (ACTRN12606000185561).

## Strengths and limitations of this study

► Large acute rehabilitation trial provides descriptive data and overview of regional service patterns.
► Little missing data provide high confidence in outcomes.
► Smaller numbers of participants and wide diversity of health services in South East Asia limit insights for this geographic region.

## INTRODUCTION

Evidence suggests stroke incidence in people of working age is on the rise.[1] In Australia, New Zealand and the UK, 25%–30% of strokes occur in people of working age (≤65 years).[2–5] With 51 000 strokes/year in Australia[2] and 152 000/year in the UK,[6] this equates to an estimated 12 750 and 38 000 strokes in people of working age each year in these countries, respectively.

With 5-year survival rates the highest for those 50 years and younger,[7] stroke survivors of working age may live many years with the physical, psychological and social consequences of stroke. Stroke survivors of working age have unique rehabilitation needs and different expectations of recovery.[8–10] Outcomes pertinent to stroke survivors of working age may include a return to independence, self-reliance and work.[11,12] Alongside the personal impact, the financial burden associated with stroke disability in people of working age, to individuals, carers and the economy is considerable. The cost of lost earnings due to reduced employment from stroke in Australia alone was estimated as US$975 million in 2013.[2] Rehabilitation

is a time limited, costly, but valued part of post stroke care.

How stroke services are organised to meet the needs of survivors varies between countries. Service priorities and guideline implementation also vary across different healthcare settings, and according to government health structures, geography and culture.[13] Acute care in organised stroke units is supported by high-level evidence, and global efforts to standardise acute medical treatments have improved stroke unit uptake, which typically includes some level of assessment and treatment by a multidisciplinary team. Intercountry or regional differences in rehabilitation access and pathways following acute care are less well understood and therefore a focus of this study.

Post acute stroke rehabilitation services often include two key models:

► Inpatient rehabilitation (IR): rehabilitation service delivered in the hospital setting.
► Community rehabilitation (CR): rehabilitation service delivered as an outpatient at a clinic or day hospital or in the patient's own home. This model includes early supported discharge (ESD) which offers early discharge from acute hospital and provision of rehabilitation and support in the home.

The AVERT (A Very Early Rehabilitation Trial) trial of very early rehabilitation[14] included 2104 participants from 5 countries (3 geographic regions; Australasia, South East (SE) Asia, UK) who were recruited within 24 hours of stroke, and randomised to two rehabilitation interventions: very early intensive mobilisation plus usual care, or usual care alone. Twelve-month disability and health-related quality of life (HRQoL) outcomes were not different between intervention groups.[15] This large dataset provided a unique opportunity to explore rehabilitation pathways and 12-month outcomes for the subgroup of participants of working age.

Specifically, our aims were:

i. To describe the demographics of the working age stroke population in three regions.
ii. To explore the distribution of these individuals across four post acute rehabilitation pathways within 3 months post stroke in three geographic regions.
iii. To describe 12-month outcomes: disability, mobility, depression, quality of life, informal care and return to work (RTW) in three regions.

## METHODS
We have defined 'working age stroke' as those aged 18–65 years who are considered to be of working age; as pension age in the UK, Australia and New Zealand is 65–67 years and minimum retirement in Malaysia and Singapore age is 60–62.[16]

### Design
This is a post hoc exploratory descriptive evaluation using data from the AVERT trial. AVERT was a single-blind,

randomised controlled trial with a comprehensive economic evaluation in which participants were recruited from 56 acute stroke units between 2006 and 2015. Eligible participants for AVERT were aged 18 years or older, with no significant premorbid disability (modified Rankin Score (mRS) ≤2), with confirmed first or recurrent stroke, admitted to a stroke unit within 24 hours of stroke onset, and medically stable. The AVERT trial method has been described in detail previously[14] with the trial protocol published.[17] Human Research Ethics Committee approval was obtained from all sites and conformed to the Declaration of Helsinki. Participants or representatives provided individual consent. We extracted patient demographics, baseline stroke characteristics, discharge location, rehabilitation care and patient outcomes for participants≤65 years and grouped these into the three geographic regions with similar health and social organisation: Australasia (Australia and New Zealand), UK (England, Scotland, Wales, Northern Ireland) and SE Asia (Singapore and Malaysia).

### Patient and public involvement
A stroke consumer provided input throughout the trial planning and protocol development, site training, trial progress as a member of the trial management and executive committees, and in communication of results. No public were involved.

### Demographics and baseline stroke characteristics
Demographics included age, sex, presence of comorbid vascular risk factors, previous stroke, premorbid disability (mRS)[18] walking independence and living arrangements prior to stroke. Baseline stroke characteristics included stroke severity (National Institute of Health Score (NIHSS) score: mild (NIHSS 1–7), moderate (8–16) and severe (>16)[19] and stroke type (Oxfordshire Community Stroke Type Project classification).[20] Baseline walking was assessed using the gait component of the Mobility Scale for Acute Stroke[21] which measures the amount of assistance and independence with walking 10 m indoors with or without gait aid using a 6-point scale ranging from 1 (unable) to 6 (independent).

### Post acute discharge pathways
Discharge to home and rehabilitation (inpatient and community) data were collected at both 3 and 12 months by face-to-face interview (blinded assessor) with participants and/or carers, with supporting documentation sourced from healthcare providers. Discharge to supported care (nursing home, supported accommodation) was not included as small numbers were expected. Our focus was on determining resources used, that is, location of care (hospital, community, home) and days of rehabilitation, rather than on the content of the therapy itself. Information about rehabilitation services collected at the 3-month follow-up was used to classify participants into four different rehabilitation pathways (figure 1).

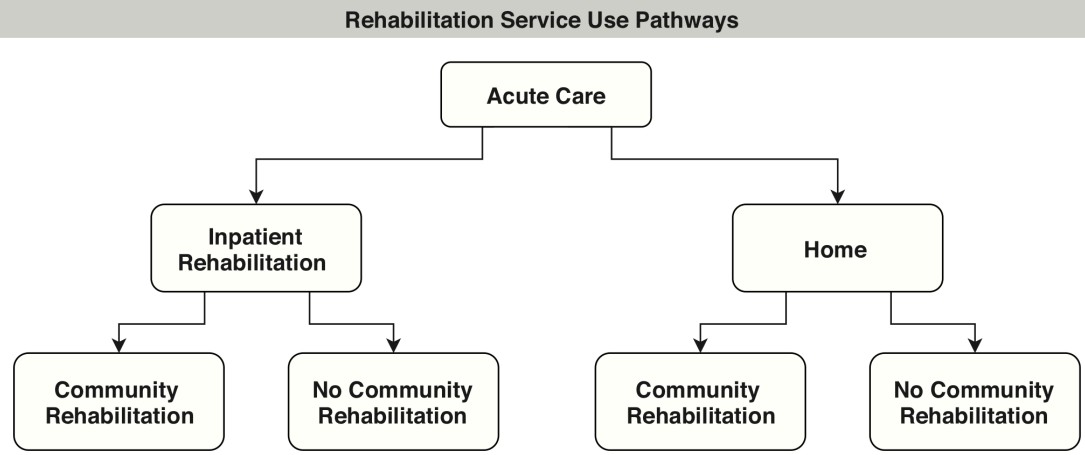

**Figure 1** Four post acute discharge pathways: inpatient rehabilitation only; inpatient rehabilitation followed by community rehabilitation; home with community rehabilitation; and home with no community rehabilitation. Note: pathway does not include discharge to supported accommodation.

## 12-month outcomes

12-month outcomes are reported for the four post acute discharge pathways. We selected 12-month outcomes which included measures aligned with the International Classification of Functioning Disability and Health Framework.[22]

### Functioning

The mRS[18] is an ordinal scale ranging from 0 (no disability) to 5 (severe disability) or 6 (death). The proportion with favourable outcome defined as mRS scores 0–2 (no or minimal disability) is reported here.

### Activity

We used the Rivermead Motor Assessment Gross Function tool (a 13-point scale[23]) reliable in stroke[24] to measure community mobility at 3 and 12 months. We report the proportion achieving a score of 10–13 at 12 months, which indicates an ability to walk independently outside the home.[25]

### Participation

Of those working prior to stroke, we report the proportion of stroke survivors who returned to work at 12 months in any capacity (part time or full time). Quality of life is reported using the Assessment of Quality of Life measure (AQoL-4D). The questionnaire comprises 15 items over 5 subscales: independent living, social relationships, physical senses, psychological well-being and illness.[26] The AQoL utility score, which refers to the value people place on their HRQoL, ranges from –0.04 (state worse than death), death (0.0) through to 1.0 (excellent).[27] We report the proportion of stroke survivors scoring *poor* or *worse than death* (score –0.04–0.4) at 12 months.

### Body function

We extracted depression data from the Irritability Depression and Anxiety (IDA) scale and report the proportion with 'Borderline' (score 4–6) or 'Morbid' depression (score >7)[28] at 12 months.

### Environmental factors

We report the proportion of stroke survivors requiring informal care at 12 months, defined as the participant requiring help with activities of daily living beyond that provided by formal support services.

### Analyses

Baseline, rehabilitation service use within 3 months and 12-month outcome data are presented using descriptive analysis (n, %) using STATA IC for all analyses. Median and IQRs are presented for age, baseline stroke severity (NIHSS score) and length of stay. We describe pathway use by baseline stroke severity (mild, moderate and severe), and by younger (18–45 years) and older (46–65) age. We elected not to conduct formal statistical comparisons given the exploratory nature of the study, the expected small sample sizes of subgroups across the rehabilitation pathways in the study and, most importantly, we had no a priori hypotheses about the relationship between rehabilitation pathways and outcomes (or any other factors) that we felt could be formally tested in a robust way in this dataset given these expected limitations.

## RESULTS

In the AVERT trial, 668 participants were ≤65 at time of stroke; 363 participants were from Australasia, 161 from the UK and 144 from the SE Asia region. Characteristics of participants are presented in table 1. The median age of participants was similar across the three regions (58 years) with a high proportion of men (73%). Prior to stroke, 99% of participants were living independently, 88% had no disability (mRS=0), 97% were walking without a gait aid and 62% were working. The highest proportion of participants with hypertension, diabetes mellitus and ischaemic heart disease was from SE Asia. Nearly two-thirds of the people in SE Asia had lacunar strokes. The UK had the highest proportion of current smokers.

**Table 1** Premorbid and baseline stroke characteristics of participants ≤65 by region

| | Australasia n=363 | UK n=161 | SE Asia n=144 | Total n=668 |
|---|---|---|---|---|
| **Baseline characteristics** | | | | |
| Age, years | 57.6 | 58.4 | 57.6 | 57.9 |
| Median (IQR) | (49.5–62.3) | (51.5–62.6) | (51.8–61.9) | (50.3–62.2) |
| Range | 19.93–65.99 | 37.53–65.98 | 16.4–65.54 | 16.4–65.99 |
| **Sex, n (%)** | | | | |
| Female | 98 (27.0) | 56 (34.8) | 50 (34.7) | 204 (30.5) |
| Male | 265 (73.0) | 105 (65.2) | 94 (65.3) | 464 (69.5) |
| **Risk factors, n (%)** | | | | |
| Hypertension | 198 (54.5) | 84 (52.2) | 105 (72.9) | 387 (57.9) |
| Hypercholesterolaemia | 119 (32.8) | 67 (41.6) | 54 (37.5) | 240 (35.9) |
| Diabetes mellitus | 68 (18.7) | 31 (19.3) | 61 (42.4) | 160 (24.0) |
| Ischaemic heart disease | 58 (16.0) | 20 (12.4) | 24 (16.7) | 102 (15.3) |
| Atrial fibrillation | 36 (9.9) | 11 (6.8) | 11 (7.6) | 58 (8.7) |
| **Smoking, n (%)** | | | | |
| Never smoked | 139 (38.3) | 51 (31.7) | 78 (54.2) | 268 (40.1) |
| Smoker* | 129 (35.5) | 70 (43.5) | 49 (34.0) | 248 (37.1) |
| Ex-smoker* | 91 (25.1) | 39 (24.2) | 16 (11.1) | 146 (21.) |
| Unknown | 4 (1.1) | 1 (<1) | 1 (<1) | 6 (<1) |
| **Premorbid data** | | | | |
| Modified Rankin scale, n (%) | | | | |
| 0 | 319 (87.9) | 133 (82.6) | 135 (93.8) | 587 (87.9) |
| 1 | 26 (7.2) | 17 (10.6) | 5 (3.5) | 48 (7.2) |
| 2 | 18 (5.0) | 11 (6.8) | 4 (2.8) | 33 (4.9) |
| Independent walking no aid, n (%) | 356 (98.1) | 151 (93.8) | 139 (96.5) | 646 (96.7) |
| **Living arrangement at time of admission, n (%)** | | | | |
| Home alone | 52 (14.3) | 38 (23.6) | 22 (15.3) | 112 (16.8) |
| Home with someone | 310 (85.4) | 122 (75.8) | 121 (84.0) | 553 (82.8) |
| Supported accommodation | 1 (<1) | 0 (0) | 1 (<1) | 2 (<1) |
| Other | 0 (0) | 1 (<1) | 0 (0) | 1 (<1) |
| **Employment status, n (%)** | | | | |
| Working at time of stroke | 238 (65.6) | 89 (55.3) | 86 (59.7) | 413 (61.8) |
| Full time | 188 (79.0) | 66 (74.2) | 73 (84.9) | 327 (79.2) |
| Part time | 50 (21.0) | 23 (25.8) | 13 (15.1) | 86 (20.8) |
| **Baseline stroke** | | | | |
| First stroke, n (%) | 315 (86.8) | 141 (87.6) | 112 (77.8) | 568 (85.0) |
| NIHSS score, median (IQR) | 6 (4–11) | 6 (4–11) | 5 (4–8) | 6 (4–10) |
| Mild (1–7) | 213 (58.7) | 93 (57.8) | 99 (68.8) | 405 (60.6) |
| Moderate (8–16) | 107 (29.5) | 56 (34.8) | 39 (27.1) | 202 (30.2) |
| Severe (>16) | 43 (11.9) | 12 (7.5) | 6 (4.2) | 61 (9.1) |
| **Stroke type, n (%)** | | | | |
| Oxfordshire Stroke Classification | | | | |
| Total anterior circulation infarct | 61 (16.8) | 28 (17.4) | 8 (5.6) | 97 (14.5) |
| Partial anterior circulation infarct | 109 (30.0) | 55 (34.2) | 21 (14.6) | 185 (27.7) |
| Posterior circulation infarct | 53 (14.6) | 11 (6.8) | 7 (4.9) | 71 (10.6) |

**Table 1** Continued

|  | Australasia n=363 | UK n=161 | SE Asia n=144 | Total n=668 |
|---|---|---|---|---|
| Lacunar infarct | 80 (22.0) | 50 (31.1) | 91 (63.2) | 221 (33.1) |
| Haemorrhage |  |  |  |  |
| Intracerebral haemorrhage | 60 (16.5) | 17 (10.6) | 17 (11.8) | 94 (14.1) |
| Baseline walking, n (%) (Mobility Scale for Acute Stroke Walking Score) |  |  |  |  |
| Independent | 45 (12.4) | 31 (19.3) | 12 (8.3) | 88 (13.2) |
| Supervised or assisted | 200 (55.1) | 61 (37.9) | 83 (57.6) | 344 (51.5) |
| Unable to walk | 117 (32.2) | 69 (42.9) | 49 (34.0) | 235 (35.2) |
| Missing or unknown | 1 (<1) | 0 (0) | 0 (0) | 1 (<1) |

*Smoker is defined as a current smoker or participant who had quit smoking in the past 2 years, and an ex-smoker as a participant who had quit smoking more than 2 years ago.
NIHSS, National Institute of Health Stroke Scale.

## Rehabilitation pathways

The pathways of rehabilitation service use to 3 months, and 12-month outcome summaries for each region are shown in figures 2–4. Of the 668 participants, 94% (n=631/668) followed one of the four common pathways of post acute discharge. Most were discharged home (n=384/668, 57%) or to IR (n=261/668, 39%). Fourteen participants died in acute care. The discharge destination of the remaining participants was a nursing home (n=5), with missing discharge (n=4) and community rehabilitation (n=14) data.

In the first 3 months post stroke, 67% of participants received rehabilitation (IR or CR) services. Only 8% were still receiving rehabilitation services at 12-month follow-up. The proportion receiving IR was higher in Australasia than other regions (Australasia 52%; UK 25%; SE Asia 23%). The length of stay in IR was longest in UK (median 35 days) and shortest in SE Asia (median 18 days) and the UK had higher participation in CR than other regions (UK 65%; Australasia 61%; SE Asia 39%). It was most common for UK stroke survivors to go home from acute care and receive CR (48%). For those discharged home from acute care, the length of stay in hospital was similar in the three regions (median 4–5 days). The number of working age survivors who went home after a short acute care stay and received no rehabilitation (IR or CR) was unexpectedly high (Australasia 25%; UK 22%; SE Asia 40%).

The proportion of stroke survivors with mild, moderate and severe stroke, grouped by younger versus older age, and their service use pathways by regions is shown in online supplementary file 1. As could be anticipated, the highest proportion of working age survivors who did not receive rehabilitation were those with mild stroke (40%, n=156/389) compared with those with moderate (12%, n=24/200) or severe (4%, n=2/52) stroke. Less expected was our finding that 38% (n=35/92) of the younger age group aged 18–45 years did not receive any rehabilitation on discharge from acute care.

## 12-month outcomes

Of the 631 working age stroke survivors included in the four discharge pathways, 2% (n=14/631) had died by 12-month follow-up. Some 70% (n=444/631) had a favourable outcome (mRS 0–2) reporting little or no disability (mRS missing data 3%, n=20/631), yet 34% (n=217/631) were still receiving informal care. Seventy-nine per cent (n=497/631) were community ambulators (Rivermead Mobility >9) but only 57% of those working prior to stroke (n=228/398) had returned to work. The proportion who had returned to work at 12 months was similar across the three regions: Australasia 61%, SE Asia 57%, UK 48%. Twenty-eight per cent (n=174/631) had an HRQoL rated as poor or worse than death: UK 34%, SE Asia 27% and Australasia 25%. Borderline or morbid depression was reported by 44% (n=276/631), with the highest proportion in the UK 52% and the lowest proportion in SE Asia 31%. The largest proportion of missing data was for depression as some stroke survivors were unable to complete the questionnaire (IDA, 10%, n=65/631).

## DISCUSSION

This report of rehabilitation service use and 12-month outcomes of working age stroke survivors across three geographic regions with differing health services using this well-characterised and large dataset has yielded some unique findings. Interestingly, while countries in all three regions in this study have universal healthcare as a core principle of their healthcare system, we saw considerable variation in rehabilitation pathways and service use across the regions. For example, around 50% of working age stroke survivors in Australasia received IR, compared with around 25% of stroke survivors in the UK or SE Asia. In the case of SE Asia, many went straight home from acute stroke care and 40% of these individuals had no further rehabilitation. Whether this reflects differences in access to IR services between these regions, or other factors, was

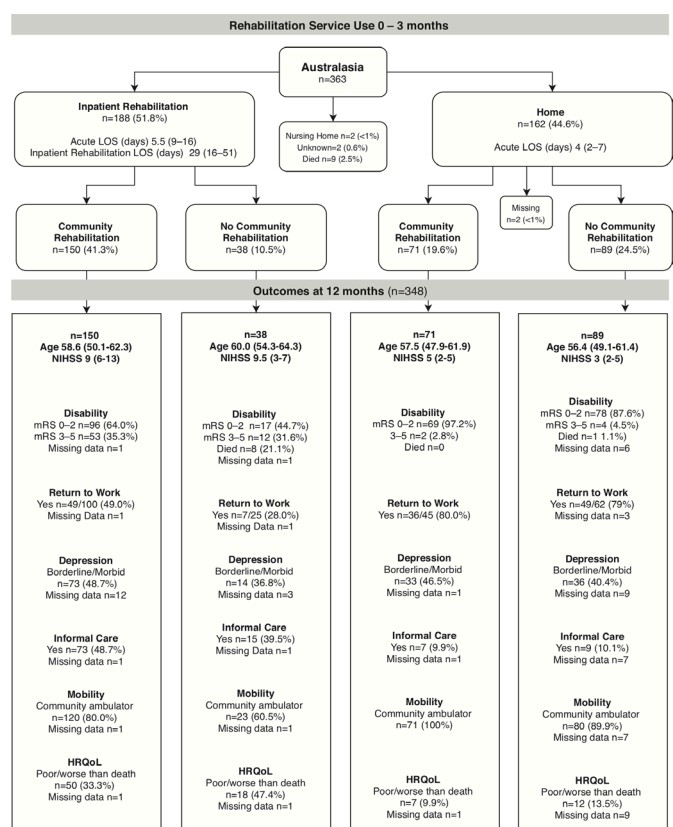

**Figure 2** Rehabilitation service use and outcomes of participants ≤65 years in Australasia. Participants in inpatient rehabilitation (IR) at 3 months n=28 (7.7%); at 12 months n=3 (<1%). Participants receiving community rehabilitation (CR) at 3 months n=1 (<1%), at 12 months n=39 (10.7%). Participants who had not been discharged from their initial acute admission or were still in IR at the time of 3-month follow-up were allocated to the appropriate pathway using CR data collected at 12 months. Length of stay (LOS), age, National Institute of Health Score (NIHSS) data are median (IQR), depression (Irritability Depression and Anxiety Scale), health-related quality of life (HRQoL; Assessment of Quality of Life measure), community ambulator defined as Rivermead Motor Assessment Gross Function>9, 12-month outcomes data are % of group, except return to work data which are % of those working prior to stroke. Missing 12-month outcomes data: unknown missing n=10/69 (14.5%); alive missing n=59/69 (85.5%). mRS, modified Rankin scale.

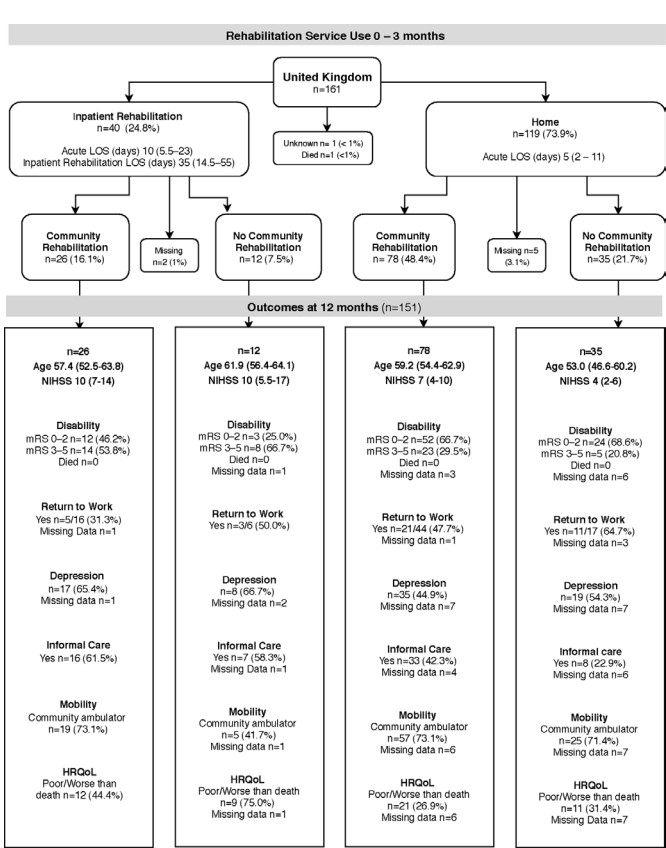

**Figure 3** Rehabilitation service use and outcomes of participants ≤65 years in UK. Participants in inpatient rehabilitation (IR) at 3 months n=8 (5.0%); at 12 months n=1 (<1%). Participants receiving community rehabilitation (CR) services at 3 months n=1 (<1%), at 12 months n=11 (6.8%). Participants who had not been discharged from their initial acute admission or were still in IR at the time of 3-month follow-up were allocated to the appropriate pathway using CR data collected at 12 months. Length of stay (LOS), age, National Institute of Health Score (NIHSS) data are median (IQR), depression (Irritability Depression and Anxiety Scale), health-related quality of life (HRQoL; Assessment of Quality of Life measure), community ambulator defined as Rivermead Motor Assessment Gross Function >9,12-month data are % of group, except return to work data which are % of participants in group working prior to stroke. Missing 12-month outcomes data: unknown missing n=15/71 (21.1%); alive missing n=56/71 (78.9%). mRS, modified Rankin scale.

not examined in this study. The SE Asia data from AVERT included data from Singapore and Malaysia. Singapore is classified as an 'advanced economy' and Malaysia as a 'developing economy'.[29] The Malaysian Clinical Practice Guidelines 2012[30] do not address care beyond the acute stroke unit. Indeed, all working age stroke survivors in SE Asia admitted to IR (n=33) were from Singapore. It should be noted that Singapore stroke guidelines[31] recommend that 'patients who had a stroke should receive organised inpatient multidisciplinary rehabilitation' (online supplementary file 2). In this dataset, only 51% (n=33/65) of participants from Singapore received IR.

The larger proportion of working age stroke survivors from SE Asia who went home with no rehabilitation

compared with those in Australasia or UK may be partly explained by a difference in stroke severity. However, median baseline stroke severity for this group was similar across the three regions (median NIHSS 5 SE Asia; 3 AUS; 4 UK). It may reflect the access to CR services in the region. There is currently no protocol in Malaysia which specifically addresses the transfer of care for long-term post stroke management at the community level.[32] A recent study of post stroke management in Malaysia[32] found that almost one-third of patients (31%) were not referred to any rehabilitation facility. They cited the lack of awareness among physicians regarding the role of neurorehabilitation and the lack of coordination of

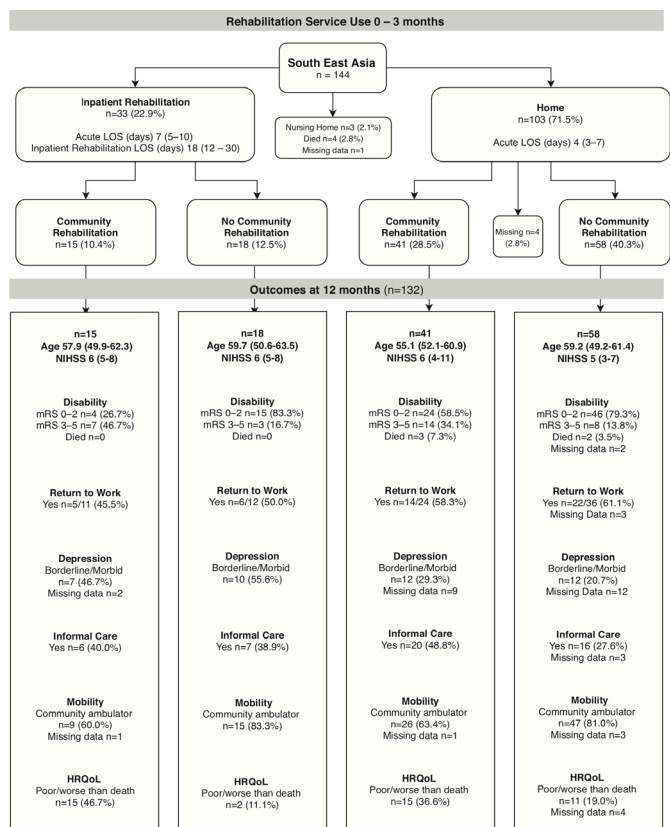

**Figure 4** Rehabilitation service use and outcomes of participants ≤65 years in South East Asia. Participants in inpatient rehabilitation (IR) at 3 months n=0; at 12 months n=0, Participants receiving community rehabilitation (CR) services at 3 months n=2 (1.4%), at 12 months n=0. Participants who had not been discharged from their initial acute admission or were still in IR at the time of 3-month follow-up were allocated to the appropriate pathway using CR data collected at 12 months. Length of stay (LOS), age, National Institute of Health Score (NIHSS) data are median (IQR), depression (Irritability Depression and Anxiety Scale), health-related quality of life (HRQoL; Assessment of Quality of Life measure), community ambulator defined as Rivermead Gross Motor Function >9, 12-month data are % of group, except return to work data which are % of participants in group working prior to stroke. Missing 12-month outcomes data: unknown missing n=6/40 (15%); alive missing n=34/40 (85%). mRS, modified Rankin scale.

post stroke care beyond the acute phase as contributing factors. It is possible that participants were offered a referral for outpatient rehabilitation services but declined due to the cost incurred, or that the group included overseas participants who were not eligible for local rehabilitation services.

In contrast with SE Asia, the lower rates of IR in the UK appear to be offset by the higher use of CR, with 48% of UK working age stroke survivors discharged home from acute care with CR. ESD services have been a focus of service development in the UK over a number of years. The ESD model has been shown to reduce length of stay and long-term dependency at 12 months in people with mild-to-moderate stroke severity.[33] We found that the majority of stroke survivors who received CR were those with mild and moderate stroke, consistent with guideline criteria for an ESD service.[34] Only 20% of survivors in Australasia went directly home with CR. Although ESD has been recommended in the Australian Clinical Practice Guidelines[35 36] since 2005 (online supplementary file 2), if we assume that episodes of CR reflect ESD, so far this model does not appear to be common practice in Australia. A challenge with interpreting CR data is that it is often poorly described, and the recommended comprehensive ESD may not be implemented.

We were surprised to find that nearly a quarter of stroke survivors in the UK and Australasia had no rehabilitation services in the first 3 months post stroke. None of these individuals had severe stroke, so perceived absence of 'rehabilitation potential' would not explain this finding. In the UK and Australasia, a large proportion (30%–47%) of the younger age group 18–45 went home with no rehabilitation. The same pattern was present for those with mild stroke at baseline, with 34%–38% of these individuals in UK and Australasia not receiving rehabilitation. This finding may relate to professionals' expectations that younger survivors and those with mild stroke will spontaneously recover without services. The need for rehabilitation services may not have been identified; 'mild' symptoms may not be apparent in the acute setting, and some, such as mood disturbance, may have a delayed onset. Stroke is a sudden, life-changing event for a young adult, the effects of which may not be anticipated during acute care. A systematic review of qualitative research reports a young adult stroke describing; 'When you're in hospital, you don't really think about how your life is going to be when you leave. You think that you'll just carry on as before, but you don't. You get home and that's when the trauma starts'.[37] A flexible model of stroke service use and follow-up is indicated, such as the 'point of contact with specialist services' recommended in Parkinson's Disease,[3] to provide ongoing support for working age stroke.

Across all three regions, we found that 12-month outcomes for those who did not receive rehabilitation were less than optimal. Whether an episode of rehabilitation would have improved these outcomes is unknown. What is perhaps most pertinent is whether the rehabilitation needs of patients in the acute phase of care were formally assessed. This was not a focus of the AVERT trial. Advances in post stroke interventions, such as thrombolysis and endovascular clot retrieval, have led to ongoing reductions in acute length of stay post stroke.[38 39] Less time in hospital may limit time for multidisciplinary assessment and identification of post discharge needs. With a median acute length of stay for participants discharged directly home of 4–5 days in each of the three regions, it may be difficult for rehabilitation needs to be adequately assessed. The need for standardised assessment of patients in acute care for stroke rehabilitation has led to the development of Assessment for Rehabilitation: Pathway and Decision-Making Tool.[40] It is now recommended that all

stroke survivors who are not for palliative care should be considered for rehabilitation.[40] Our data suggest that for a significant number of working age stroke survivors, the acute care stay may be their only contact with specialised medical and allied health stroke services, and that those under 45 years and those with mild stroke may be vulnerable to a lack of coordinated specialist post discharge care.

In this study, 70% reported little to no disability at 12 months (favourable outcome mRS 0–2), which is consistent with previous research of working age stroke survivors.[41–43] A large number of working age stroke survivors reported low mood (44% with borderline or morbid depression) and poor quality of life (28% rated HRQoL as poor or worse than death; missing data n=30). Only 7% of healthy people the same age report this level of HRQoL,[27] suggesting the need for research to determine if targeted interventions can improve QoL in working age stroke survivors. Depression has been shown to be independently associated with low HRQoL in working age stroke survivors[44] and may be improved with psychosocial, neuropsychological or pharmacological interventions.[5] The Australian Stroke Guidelines[45] recommend routine assessment of mood disturbance post stroke, and the UK[5] has identified that service improvements in the organisation and delivery of psychological services for stroke survivors are needed. The psychosocial needs of working age stroke survivors are commonly reported as being 'unmet'[46–48] and surveillance and rehabilitation programmes that specifically address these issues appear needed.

The proportion of working age stroke survivors (57%) in this study who had returned to work at 12 months was low but consistent with previous research.[49 50] Poor rates of RTW despite good functional outcomes have been reported. For example, Varona et al[43] evaluated long-term outcomes in younger adults after first ischaemic stroke and found while 90% were independent, only 53% had returned to work. RTW post stroke is complex and multifactorial,[51 52] with optimal interventions to improve RTW outcomes for people of working age poorly understood.[53] A recent systematic review[50] concluded that either specialised vocational rehabilitation, conventional stroke rehabilitation or their combination is needed to increase return-to-work rates and improve the quality of life for stroke survivors of working age.

### Study limitations

We elected to explore data for all stroke survivors of 'working age' (ie, ≤65 years) which is consistent with other working age definitions. However, in our study population, only 62% of participants were in formal paid employment prior to their stroke. We acknowledge that some people continue to work after age 65, and that the absence of information about unpaid, productive work is a limitation of the data. Further, the absence of information about the specific goals or types of rehabilitation has limited our ability to describe in more detail the

rehabilitation services on offer. We relied on clinicians, patients and families reporting of rehabilitation services, and did not gather detailed information as part of the trial about services (such as professions involved, details of service models) that could help benchmark services to any common standard. Such information would be valuable in future research. Smaller numbers of participants and wide diversity of health services in SE Asia limit insights for this region. Nevertheless, the data available from three separate geographic regions offer unique insights into who gets into rehabilitation, and what type, but not why. We elected not to run exploratory analyses on these data given the very small numbers of patients in subgroups of interest, such as those of younger age (<45 years), different genders or with mild stroke. Finally, younger stroke survivors live many years with post stroke disability and their needs may change over time. This study reflects only the first 12 months after stroke.

### CONCLUSIONS

Despite the majority of working age stroke survivors in this study receiving rehabilitation services within the first 3 months, many reported low mood, poor QoL and failure to RTW at 12 months. Over a quarter (28%) of working age stroke survivors received no rehabilitation services. Younger individuals and those with mild stroke may not be offered rehabilitation yet report suboptimal outcomes in the longer term. At present, there is little research targeting the needs of younger survivors or indicating benefit; this is an avenue for further enquiry. Further research into the most effective and cost-effective rehabilitation service use pathway for young adults with varying severity of stroke is needed to assist in the development of specific evidence-based guidelines to inform service delivery.

**Author affiliations**
[1]Physiotherapy, Royal Free London NHS Foundation Trust, London, UK
[2]Stroke, Florey Institute of Neuroscience and Mental Health, Heidelberg, Victoria, Australia
[3]Academic Section of Geriatric Medicine, Institute of Cardiovascular and Medical Sciences, University of Glasgow, Glasgow, UK
[4]Physiotherapy, Hospital Canselor Tuanku Muhriz, Pusat Perubatan UKM, Cheras, Kuala Lumpur, Malaysia
[5]Faculty of Health, Art and Design, Swinburne University of Technology, Hawthorn, Victoria, Australia
[6]Department of Physiotherapy, Singapore General Hospital, Singapore
[7]Deakin Health Economics, Deakin University, Burwood, Victoria, Australia

**Acknowledgements** The authors thank the AVERT Collaboration investigators for their hard work and dedication, and the participants in this trial and their supporters. They also thank Brooke Parsons, Stroke Consumer, for her important and sustained contribution to this study. The Florey Institute of Neuroscience and Mental Health acknowledges the support received from the Victorian Government via the Operational Infrastructure Support Scheme.

**Collaborators** AVERT Trialists' Collaboration.

**Contributors** RW and JB conceived the study and all authors contributed to the study design. Rehabilitation services data were devised by MM, data were managed by JMC and collected by the AVERT Trialists' Collaboration. RW performed

data analysis. Interpretation was provided by JB, JMC, LBC, PL, MAK and DT. RW was the primary writer of the manuscript, with revisions and intellectual content from all authors. All authors provided final approval of the version to be published.

**Funding** The AVERT trial was initially supported by the National Health and Medical Research Council (NHMRC) of Australia (grant numbers 386201 and 1041401). Additional funding was received from Chest Heart and Stroke Scotland (Res08/A114), Northern Ireland Chest Heart and Stroke, Singapore Health (SHF/FG401P/2008), the UK Stroke Association (TSA2009/09) and the UK National Institute of Health Research (HTA Project 12/01/16). NHMRC fellowship funding was provided to JB (1058635, 1154904). JB also received fellowship funding from the Australia Research Council (0991086) and the National Heart Foundation. RW received a Clinical Stipend from the NHMRC Centre of Research Excellence in Stroke Rehabilitation and Brain Recovery (1077898).

**Competing interests** RW reports a clinical stipend from a NHMRC grant. PL reports grants from NIHR, Stroke Association, Chest Heart and Stroke Scotland. JB reports grants from the NHMRC, Stroke Association, Chest Heart and Stroke Scotland, Northern Ireland Chest Heart and Stroke, NIHR, ARC and NHF. MM reports grants from NHMRC. DT and JB reports grants from Singapore Health. JMC, LBC and MAK report nothing to disclose.

**Patient consent for publication** Not required.

**Provenance and peer review** Not commissioned; externally peer reviewed.

**Data availability statement** Data are available upon reasonable request. Data may be shared by writing to the corresponding author to obtain details of procedures and processes.

**ORCID iDs**
Rosy Walters http://orcid.org/0000-0002-3927-2666
Janice M Collier http://orcid.org/0000-0003-2950-4870
Julie Bernhardt http://orcid.org/0000-0002-2787-8484

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
