## [Reviewer comments · BMJ Open]

ARTICLE DETAILS

TITLE (PROVISIONAL)	Exploring post-acute rehabilitation service use and outcomes for working age stroke survivors (≤ 65 years) in Australia, United Kingdom and South East Asia: Data from the international AVERT trial
AUTHORS	Walters, Rosy; Collier, Janice M.; Braighi Carvalho, Lillian; Langhorne, Peter; katijjahbe, Md Ali; Tan, Dawn; Moodie, Marj; Bernhardt, Julie

VERSION 1 – REVIEW

REVIEWER	Simone Dorsch Australian Catholic University Australia
REVIEW RETURNED	06-Jan-2020

GENERAL COMMENTS	General comments:  • On the whole very well written and easy to read • Overuse of abbreviations makes the manuscript more difficult to read – particularly in the Abstract • Objectives not clear - primarily to describe outcomes for younger stroke survivors (as stated in abstract) or primarily to describe delivery of rehab or to describe differences across regions (as stated in Intro) • Malaysia and Singapore are the only two countries in the south-east asia region that are in the study - if this is the case, this does not seem to be representative enough to be called SE Asia and as these two countries are very different economically and in delivery of health services combining their data may not be helpful. There are also significant differences in delivery of rehabilitation in Malaysia which I believe has no inpatient rehabilitation units, making the comparison of rehab service delivery between other countries and Malaysia not very helpful • As this is a descriptive study rather than a RCT – would it be more applicable to use the STROBE statement than the CONSORT statement Abstract  • Lines 5-10; the manuscript does not appear to be written with the aim of providing information to young stroke survivors – the Objectives section of the Abstract would be clearer if it was consistent with the Objectives stated at the end of the Intro. It would also be clearer to refer to the large data-set obtained from AVERT and to describe the purpose and outcomes of AVERT elsewhere • I am not sure that it is accurate to describe the interventions as though this is an intervention study as it is not concerned with the interventions or outcomes of the AVERT trial but with other uses of the data-set
---

	 • As the results section of the Abstract is concerned with the demographics and the overall outcomes not the between group outcomes of AVERT it is confusing to have described the intervention arms of the AVERT study Intro The Intro is very comprehensive and provides the necessary background to the study  • Page 6, lines 29-32; does the word 'regions' refer to geographical regions or something else Methods  • Is there any data about the use of formal care – it would be good to report this as well as informal care? • I don't think it makes sense to combine data from Malaysia and Singapore when these countries have very different health care Results On the whole very clearly written  • Lines 25-31; it is not clear why this selection of results is reported in text Discussion This is a very comprehensive and pertinent discussion of the implications of this data  • Lines 20-26; my understanding of stroke rehabilitation in Malaysia is that there are no inpatient rehabilitation units as exist in UK, Singapore and Australia - and very little outpatient services - this has led to the founding of NASAM which is an NGO supplying stroke rehabilitation outpatient services to fill this gap
--	--

REVIEWER	Richard Bohannon Campbell University, USA
REVIEW RETURNED	07-Jan-2020

GENERAL COMMENTS	1) In the Abstract the authors refer to "(iii) 12 month outcomes; disability..." Disability and such are outcomes. Do the authors mean to use a colon rather than semicolon after "outcomes?" 2) There is a lot of thoughtful discussion of the findings in this paper, but I don't see them having much of an affect on practice or outcomes.
--

REVIEWER	Desirée Valera-Gran Miguel Hernández University, Spain
REVIEW RETURNED	13-Jan-2020

GENERAL COMMENTS	The manuscript titled "Post-acute rehabilitation service use and outcomes for working age stroke survivors (in Australia, United Kingdom and South East Asia: Data from the international AVERT trial" is a well-written article about an important health issue from a novel perspective. However, there are some concerns that the authors should address before considering the manuscript for publication.  1. The authors provided quality and accurate data from a randomized trial conducted in stroke survivors recruited in an acute phase and followed during at least 12 months after rehabilitation. They indicated that participants were randomly
--

	distributed in two groups of rehabilitation intervention (very early intensive mobilization+usual care and usual care alone, page 6, line 10). However, they mentioned that the participants were distributed across four post-acute rehabilitation pathways. Moreover, according to that displayed in Figure 1, the groups were classified as “Inpatient rehabilitation” and “Home”. This is quite unclear and it is difficult to see what type of intervention was made in each group and which groups finally were compared.  2. In the section “12 month outcomes” of the Methods (page 8, lines 4-9), the authors indicated that they included measures from each domain of The ICF. Given that this is the main outcome of the present study, it would be of valuable interest to add information about the variables collected. 3. Table 1 requires some changes: 1) P-value should be calculated to analyze the differences between the study region; 2) the column with the estimates for the total sample should be located in the first position, i.e. before the columns of study regions. However, it would be more interesting to present the participant characteristics by the rehabilitation intervention group. 4. To improve the reading of figure 1, the authors should include n and %. 5. If the results are presented by rehabilitation groups, Figures 2, 3, and 4 should be included as supplementary material. 6. Although analyses concerning 12 month outcomes were performed by region, it would be more interesting to analyze data by type of intervention. 7. Perhaps there is a typo in age category (16-45 instead of 18-45) in the table Supplement 1. 8. The present study only displays results of a descriptive analysis. Given that this study yield quality and accurate data, it would be more interesting explore the association between the rehabilitation interventions and 12 outcomes adjusting for confounding study covariates, for instance region, age, etc. This would provide more useful and interesting results and reinforce the study implications and conclusions.
--	--

REVIEWER	David Conradsson Division of Physiotherapy, Department of Neurobiology, Care Sciences and Society, Karolinska Institutet, Sweden
REVIEW RETURNED	24-Jan-2020

GENERAL COMMENTS	Thank you for the opportunity to review this paper; a secondary analysis of the AVERT trial seeking to describe baseline demographics, rehabilitation pathways and outcomes 12-months in persons suffered from stroke in Australasia, South East Asia and United Kingdom. More specifically, demographics and outcomes were described for 668 stroke survivors of working age following different rehabilitation pathways (i.e. provision of inpatient/community rehabilitation, or no rehabilitation). Key findings were; irrespective of setting, that many stroke survivors reported
---

poor outcomes 12-months after stroke, especially for mood, quality of life and return to work, and about one third reported no rehabilitation services at all following acute care at stroke unit. There was also large variation in rehabilitation pathways between the 4 regions.

This is a well written paper which will make an important contribution to the field of stroke rehabilitation and health policy-makers highlighting the unmet need to provide appropriate services to address sustained disability post-stroke. Next to my enthusiasm, I have some questions which I would like the authors to clarify.

Major comments

1. Was this a pre-planned secondary analysis of the AVERT trial or an explorative study which arose from the results of the AVERT trial. In the light of transparency, I think this could be clearly stated in the paper, and included in the title and abstract.

2. While a descriptive approach is sound in relation to the research question raised in this paper, have the authors considering to also model the relation between personal factors, rehabilitation trajectories and setting (e.g. geographical region and urban/rural) in relation to outcomes at 12-months? The authors state that the descriptive approach was selected due to small sample size, but is not the current sample sufficient to perform robust statistical models beyond descriptive statistics?

3. While people of working age is an important subgroup with regards to rehabilitation services and outcomes post-stroke, I do not see the merits of excluding people above 65 years of age in this paper. For instance, there is a possibility that younger patients may be given higher priority for rehabilitation than older patients. Such notion could have been addressed in this paper if older adults were not excluded. Please justify.

Specific comments

Introduction

1. The introduction is nicely written and introduce the reader to the study rationale and aim. On page 6 (line 5-19), I would suggest to add a richer description on the different settings and health care systems in the 4 geographical regions, including the clinical guidelines and criteria for post-stroke rehabilitation (i.e. brief description from supplement material 2).

Methods

2. Page 7, lines 52-55: "Rehabilitation data collected at 3 months were used to best represent service delivery relating to the initial stroke event." The meaning of this sentence could be phrased more explicit, such as information about rehabilitation services collected at the 3-months follow-up were used to classify participants into different rehabilitation trajectories.

3. Page 7, lines 55-57. In line with comment 2, with regards to following sentence; "Four post-acute discharge pathways, which include details of rehabilitation service use, were defined a priori". To me, stating that "details of rehabilitation services used" were collected and in the same paragraph write that "no information were collected about content of rehabilitation" is a bit contradictory. I would suggest to rephrase to improve consistency.

	4. Page 7, lines 52-57: My experience from another setting (Sweden) is that the “early-discharge-to-home” trajectory is most common for mild stroke survivors, still the rehabilitation services provided within this trajectory differs substantially between geographical regions and even in the same region/metropolitan area. The in-patient rehabilitation trajectory is from my understanding often more heterogenous regarding rehabilitation services. I would therefore suggest that the authors: 1) add a description or criteria regarding the minimal requirement in terms of rehabilitation services (e.g. type of rehab, professionals involved, duration etc.) to qualify as in-patient and community rehabilitation, and 2) if available, add data regarding the rehabilitation services provided for the different trajectories in each region. If this data is unavailable, please make a more explicit statement in the discussion that this is a limitation to this paper. Results 5. The results are clearly presented- the text is to the point and easy to follow. While Figure 2-4 provides a clear description about the baseline demographics, rehabilitation pathways and outcomes for each region separately – these figures are not so helpful for comparison between regions. Therefore, my suggestion is to subdivide the figures based on the 3 main domains of this paper; rehabilitation trajectories, baseline demographics and outcomes at 12-months, instead of presenting these domains together and split the regions. Table 1 6. To improve clarity, I would suggest to remove the “%” signs from the second to fifth column in the table and instead indicate that the information are numbers and percentages in left hand column. Discussion Relevant and well-structured. No comments.
--	--

VERSION 1 – AUTHOR RESPONSE

Reviewer’s Comments:

Reviewer: 1
Reviewer Name: Simone Dorsch
Institution and Country:
Australian Catholic University
Australia

1) Overuse of abbreviations makes the manuscript more difficult to read – particularly in the Abstract

- ➔ Thank you, we have removed most abbreviations except for region which makes the abstract much more legible. In the main text, we have used common conventions for abbreviations and restricted their use where possible.

2) Objectives not clear - primarily to describe outcomes for younger stroke survivors (as stated in abstract) or primarily to describe delivery of rehab or to describe differences across regions (as stated in Intro

- Thank you for pointing out these inconsistencies. We have reviewed the objectives and amended these in the Abstract (Page 2, Objectives) and Introduction (Page 7, Aims) to ensure they align.

3) Malaysia and Singapore are the only two countries in the south-east asia region that are in the study - if this is the case, this does not seem to be representative enough to be called SE Asia and as these two countries are very different economically and in delivery of health services combining their data may not be helpful. There are also significant differences in delivery of rehabilitation in Malaysia which I believe has no inpatient rehabilitation units, making the comparison of rehab service delivery between other countries and Malaysia not very helpful

- We readily acknowledge that the two countries included in this study are not reflective of SE Asia as a whole. In light of your comments, we have addressed this as a limitation of the study (Article Summary, page 5), and have added a comment in the Limitations section of the discussion (Discussion, Limitations, page 18.)
- The countries (Malaysia and Singapore) were clustered into geographical regions during the AVERT trial and have been reported as the SE Asia in all previous papers. For consistency with the original trial and trial reports, we have retained the regional clusters and title. We also acknowledge that the two countries differ in terms of their economics and health services available. We raised this issue ourselves in the discussion of the paper (Discussion, paragraph 1 and 2. The detail provided by our co-authors, see point below, including references to clinical guidelines (Supplement 2) serve to highlight and explore these differences.
- Co-authors Katijjahbe and Tan from the AVERT trial sites in Malaysia and Singapore contributed valuable information and feedback incorporated into the discussion and we feel confident that the information there is an accurate reflection of the availability of resources within these countries.

4) As this is a descriptive study rather than a RCT – would it be more applicable to use the STROBE statement than the CONSORT statement

- Thank you. The descriptive analysis of the data for this paper uses data gathered from AVERT, a randomised clinical trial. We agree that use of the STROBE reporting guidelines are applicable given that we have used this data as a cohort group for this paper.

Abstract

5) Lines 5-10; the manuscript does not appear to be written with the aim of providing information to young stroke survivors – the Objectives section of the Abstract would be clearer if it was consistent with the Objectives stated at the end of the Intro. It would also be clearer to refer to the large data-set obtained from AVERT and to describe the purpose and outcomes of AVERT elsewhere

- Thanks for identifying this issue regarding the consistency of objectives. Please refer to our responses (Reviewer 1), Comment 2.

6) I am not sure that it is accurate to describe the interventions as though this is an intervention study as it is not concerned with the interventions or outcomes of the AVERT trial but with other uses of the data-set. As the results section of the Abstract is concerned with the demographics and the overall outcomes not the between group outcomes of AVERT it is confusing to have described the intervention arms of the AVERT study

We agree and have made revisions and reference to this has now been removed in the abstract (which was confusing).

Intro

The Intro is very comprehensive and provides the necessary background to the study. 7) Page 6, lines 29-32; does the word 'regions' refer to geographical regions or something else

The word 'regions' does refer to geographical regions. For clarity, we have added the word 'geographical' for clarity throughout the paper.

Methods

8) Is there any data about the use of formal care – it would be good to report this as well as informal care?

The amount of formal care participants received was collected for the economic analysis in the AVERT trial and has already been reported elsewhere (Gao, L., et al. 2019 Economic evaluation of a phase III randomised controlled trial of very early mobilisation after stroke (AVERT). BMJ Open 9: e026230). For this paper, we elected to focus on the informal support provided by family, friends and carers which show the additional burden on the family unit.

9) I don't think it makes sense to combine data from Malaysia and Singapore when these countries have very different health care

We appreciate your view, but for reasons already outlined above, we have elected to retain the combined reporting of data from these countries which are more similar to each other than they are to the UK or Australia/NZ. Please refer to our responses (Reviewer 1), Comment 3.

We openly share our data on request and currently have two groups working on separate research questions that explore data from Malaysia and Singapore. We are happy to provide individual country data for future between-country analyses should that be requested.

Results

On the whole very clearly written.

10) Lines 25-31; it is not clear why this selection of results is reported in text

- We are not clear about what the reviewer was asking here. We have reviewed the selected text and still feel it is appropriate information at this point in the manuscript and does not duplicate information from tables.

Discussion

This is a very comprehensive and pertinent discussion of the implications of this data

11) Lines 20-26; my understanding of stroke rehabilitation in Malaysia is that there are no inpatient rehabilitation units as exist in UK, Singapore and Australia - and very little outpatient services - this has led to the founding of NASAM which is an NGO supplying stroke rehabilitation outpatient services to fill this gap

- Thank you, we are glad you found the discussion comprehensive.
- We recognise the differences in service provision in the different countries, which we detailed in the discussion (see response to earlier query). We also included country specific clinical guidelines to highlight the differences in service provision (Supplement Material 2). Co-authors Katijahbe and Tan from the AVERT trial sites in Malaysia and Singapore provided specific feedback and contribution to the discussion to ensure accurate reflection of availability of resources within the countries.

Reviewer: 2

Reviewer Name: Richard Bohannon

Institution and Country: Campbell University, USA

1) In the Abstract the authors refer to “(iii) 12 month outcomes; disability....” Disability and such are outcomes. Do the authors mean to use a colon rather than semicolon after “outcomes?”

→ Thank you for identifying this error. We have changed the colon to a semicolon.

2) There is a lot of thoughtful discussion of the findings in this paper, but I don't see them having much of an affect on practice or outcomes.

→ We thank you for your comments. Indeed, we fully agree. It would be unlikely that a report like this would change practice, however we started this exploration not knowing what rehabilitation pathways looked like for people with stroke because so few data of this kind exist. We hope that by presenting these data and outlining regional differences in rehabilitation service use and outcomes, we have provided data that will start conversations and may stimulate future rehabilitation research questions about the challenges facing working age people with stroke.

Reviewer: 3

Reviewer Name: Desirée Valera-Gran

Institution and Country: Miguel Hernández University, Spain

1) The authors provided quality and accurate data from a randomized trial conducted in stroke survivors recruited in an acute phase and followed during at least 12 months after rehabilitation. They indicated that participants were randomly distributed in two groups of rehabilitation intervention (very early intensive mobilization+usual care and usual care alone, page 6, line 10). However, they mentioned that the participants were distributed across four post-acute rehabilitation pathways. Moreover, according to that displayed in Figure 1, the groups were classified as “Inpatient rehabilitation” and “Home”. This is quite unclear and it is difficult to see what type of intervention was made in each group and which groups finally were compared.

→ We apologise. The exploratory nature of this paper was not made entirely clear. As reviewer 1 pointed out, including an “Intervention” label in the abstract made the rest of the approach for the study confusing. This has now been removed. We believe it is much clearer that this was a post-hoc exploration of data from the AVERT trial (a largRCT previously reported in The Lancet). This is now clearly stated in the abstract and start of methods (design).

2) In the section “12 month outcomes” of the Methods (page 8, lines 4-9), the authors indicated that they included measures from each domain of The ICF. Given that this is the main outcome of the present study, it would be of valuable interest to add information about the variables collected.

→ We have reviewed the information provided for each of the 12 month outcomes collected, and the information we have provided regarding the scope of the measures, their cut-points and their use in stroke.

→ We have altered the sub-headings of the seven 12 month outcomes to include their domain in the ICF (Method, 12 month outcomes).

3) Table 1 requires some changes: 1) P-value should be calculated to analyze the differences between the study region; 2) the column with the estimates for the total sample should be located in the first position, i.e. before the columns of study regions. However, it would be

more interesting to present the participant characteristics by the rehabilitation intervention group.

- We have clarified the objectives of the study in the abstract and introduction. Please refer to responses to Reviewer 1, Comments 5 and 6.
- We talked extensively about whether any statistical analysis of the data for this paper was warranted, given the descriptive nature of the questions we have asked in the objectives of this paper. We consulted our AVERT trial statistician Professor Leonid Churilov, whose view was that the approach we have taken is robust.
- We have added the following statement to the Analysis section of the paper “We elected not to conduct formal statistical comparisons given the exploratory nature of the study, the expected small sample sizes of subgroups across the rehabilitation pathways in the study and, most importantly, we had no a priori hypotheses about the relationship between rehabilitation pathways and outcomes (or any other factors) that we felt could be formally tested in a robust way in this dataset given these expected limitations.”

4) To improve the reading of figure 1, the authors should include n and %. If the results are presented by rehabilitation groups, Figures 2, 3, and 4 should be included as supplementary material.

- Figure 1 is not a presentation of data and is, instead, used to illustrate the four rehabilitation pathways defined in the paper and to help orient the reader to what is to follow.
- Figures 2-4 are presented as Figures rather than Supplementary Material as they present the data concerning two of the main aims/objectives of the paper ((i) the distribution of individuals across four post-acute rehabilitation pathways in the three regions; ii) the 12 month outcomes: disability, community mobility, depression, quality of life, informal care and return to work (RTW) in the three regions. We have elected to retain the current format of the paper.

5) Although analyses concerning 12 month outcomes were performed by region, it would be more interesting to analyze data by type of intervention.

- Please refer to the response to Reviewer 3, Comment 1.

6) Perhaps there is a typo in age category (16-45 instead of 18-45) in the table Supplement 1.

- Thank you for highlighting this. We have changed the age category to 18-45 in Table supplement 1

7) The present study only displays results of a descriptive analysis. Given that this study yield quality and accurate data, it would be more interesting explore the association between the rehabilitation interventions and 12 outcomes adjusting for confounding study covariates, for instance region, age, etc. This would provide more useful and interesting results and reinforce the study implications and conclusions.

- Even though this data set is large, relative to many others, the numbers are insufficient for exploring the rehabilitation dose effects with multiple adjustments for the different co-variates. In particular, the SE Asia geographical group (n=144) was sufficiently small to limit these type of analyses. The AVERT trial was powered to examine one primary outcome at 3 months (mRS). We didn't know what we would find on exploring these data, and had no a priori hypotheses that we wanted to test. We did expect to find smaller and smaller numbers of participants as we moved across regions and pathways – see the added note about Analysis in our response to Reviewer 3, comment 3 above.

Reviewer: 4

Reviewer Name: David Conradsson

Institution and Country:

Division of Physiotherapy, Department of Neurobiology, Care Sciences and

1. Was this a pre-planned secondary analysis of the AVERT trial or an explorative study which arose from the results of the AVERT trial. In the light of transparency, I think this could be clearly stated in the paper, and included in the title and abstract.

- We thank you for highlighting this further. We have stated in both the abstract and the methods section that the study is a post-hoc exploratory study (see abstract, design etc). We have also made the exploratory nature of the analysis clearer by altering the title to include, 'Exploring'

2. While a descriptive approach is sound in relation to the research question raised in this paper, have the authors considering to also model the relation between personal factors, rehabilitation trajectories and setting (e.g. geographical region and urban/rural) in relation to outcomes at 12-months? The authors state that the descriptive approach was selected due to small sample size, but is not the current sample sufficient to perform robust statistical models beyond descriptive statistics?

- Please refer to response to Reviewer 3, Comments 3 and 7.

3. While people of working age is an important subgroup with regards to rehabilitation services and outcomes post-stroke, I do not see the merits of excluding people above 65 years of age in this paper. For instance, there is a possibility that younger patients may be given higher priority for rehabilitation than older patients. Such notion could have been addressed in this paper if older adults were not excluded. Please justify.

- We selected the data for this paper to include those 65 years or younger, due to a genuine interest in exploring the younger cohort and to understand their specific post-acute journey of care and 12 month outcomes. The objectives of this paper were outlined in the context of other recent work exploring unmet needs of younger stroke survivors. These people have specific needs and outcomes which are different to older stroke survivors.
- We recognise the importance of presenting the data on the >65 years and have addressed the rehabilitation service use and outcomes of this age group in a different paper. We wanted to avoid comparing the two groups due to their different needs.

Introduction

1. The introduction is nicely written and introduce the reader to the study rationale and aim. On page 6 (line 5-19), I would suggest to add a richer description on the different settings and health care systems in the 4 geographical regions, including the clinical guidelines and criteria for post-stroke rehabilitation (i.e. brief description from supplement material 2).

- Thank you for your feedback. We feel that including detailed information on the different settings and health care systems would be beyond the scope of the introduction. We have addressed this in detail in the discussion and Supplementary Material 2.

Methods

2. Page 7, lines 52-55: "Rehabilitation data collected at 3 months were used to best represent service delivery relating to the initial stroke event." The meaning of this sentence could be phrased more explicit, such as information about rehabilitation services collected at the 3-months follow-up were used to classify participants into different rehabilitation trajectories.

Thank you for your suggested wording which improves the clarity of the methodology we used. We have changed our original wording as suggested (Post-acute discharge pathways, Page 9) "Information about rehabilitation services collected at the 3 month follow-up were used to classify participants into four different rehabilitation pathways."

3. Page 7, lines 55-57. In line with comment 2, with regards to following sentence; “Four post-acute discharge pathways, which include details of rehabilitation service use, were defined a priori”. To me, stating that “details of rehabilitation services used” were collected and in the same paragraph write that “no information were collected about content of rehabilitation” is a bit contradictory. I would suggest to rephrase to improve consistency.

- Thank you for your suggestion. We have addressed the structure and content of these lines to reflect the fact that we defined the post-acute discharge pathways using rehabilitation service use data, but that details of the type of therapy, such as which disciplines of therapy were involved, or what people were doing within their therapy sessions were not included (Page 9, Post acute discharge pathways). The limitation to describe in more detail the rehabilitation services on offer is acknowledged as one of the study limitations (Page 18, Limitations)

4. Page 7, lines 52-57: My experience from another setting (Sweden) is that the “early-discharge-to-home” trajectory is most common for mild stroke survivors, still the rehabilitation services provided within this trajectory differs substantially between geographical regions and even in the same region/metropolitan area. The in-patient rehabilitation trajectory is from my understanding often more heterogenous regarding rehabilitation services. I would therefore suggest that the authors: 1) add a description or criteria regarding the minimal requirement in terms of rehabilitation services (e.g. type of rehab, professionals involved, duration etc.) to qualify as in-patient and community rehabilitation, and 2) if available, add data regarding the rehabilitation services provided for the different trajectories in each region. If this data is unavailable, please make a more explicit statement in the discussion that this is a limitation to this paper.

- Thank you for your thoughtful comments. The focus of this paper was on the regional differences in post-acute rehabilitation in broad terms - Community or Inpatient - rather than the specifics of rehabilitation. As you have mentioned, these may differ significantly within, as well as between, regions. Although Early Supported Discharge services have defined criteria, services remain highly variable and there is no ‘one version’ of ESD, which makes defining community rehabilitation services challenging. We have included the following in the Discussion, Limitations.
- “We relied on clinicians, patients and families reporting of rehabilitation services, and did not gather detailed information as part of the trial about services (such as professions involved, details of service models) that could help benchmark services to any common standard. Such information would be valuable in future research. Smaller numbers of participants and wide diversity of health services in SE Asia limit insights for this region.”
- The AVERT study included a strong cost effectiveness evaluation that required information about services to help cost out those services. For the study, we relied on the health providers at each site to nominate what kind of rehabilitation service was offered. There was no formal definition of what constituted a ‘service’ in the eyes of the user. We agree that this information would be particularly valuable for benchmarking across different hospitals within a country and between countries. It was not in scope of the original trial and so we are unable to provide that level of detail. We can, we believe, feel confident that when clinicians, patients and families reported at the face to face follow ups about ‘rehabilitation’ that those services were targeted at their ongoing recovery.

Results

5. The results are clearly presented- the text is to the point and easy to follow. While Figure 2-4 provides a clear description about the baseline demographics, rehabilitation pathways and outcomes for each region separately – these figures are not so helpful for comparison between regions. Therefore, my suggestion is to subdivide the figures based on the 3 main domains of this paper; rehabilitation trajectories, baseline demographics and outcomes at 12-months, instead of presenting these domains together and split the regions.

→ We are glad that you found the results to be presented clearly. We did explore many options to present the data within the three figures in such a way that people could look at the data within each region and then across each region for comparison. One of those options included presenting data as you suggested, however feedback from our group and other reviewers we consulted with indicated that we lost too much data that was valuable within a region taking this approach and it became more confusing to interpret the data. Furthermore, we had feedback from stroke survivors and health professionals from the different regions that they wanted to review the journey of post-acute care through to 12 month outcomes relevant to their region. Ideally, we would include all the data within one figure, but we tried that too and it was not workable. In the end, we have elected to retain the information as presented, but we thank you for the suggestion.

Table 1

6. To improve clarity, I would suggest to remove the “%” signs from the second to fifth column in the table and instead indicate that the information are numbers and percentages in left hand column.

→ Thank you for your suggestion. We have amended the table as you have suggested

VERSION 2 – REVIEW

REVIEWER	Simone Dorsch Australian Catholic University
REVIEW RETURNED	07-Apr-2020

GENERAL COMMENTS	I am satisfied that the authors have addressed comments given in the previous review and I believe that the publication provides valuable information about outcomes in this younger age group of stroke survivors. Some minor edits as follows: Abstract; line 31 - '88% had no? disability' Strengths and Limitations of this study; - this would be easier to read if expanded into full sentences - if not word limited..... Intro; line 23 - 'alongside the personal impact....' Methods; page 8, line 48 - '?there was no public involvement' page 9, line 26 - 'information.....was used....' Discussion; page 16 - line 19 - 'for working age stroke survivors'
--

REVIEWER	Desirée Valera-Gran Miguel Hernández University, Spain
REVIEW RETURNED	23-Mar-2020

GENERAL COMMENTS	The authors have addressed properly all the issues required by the reviewers. They have also provided critical comments to answer all the queries. All the changes made in the text have improved the manuscript considerable and it is suitable for publication in the current form.
---